# Appendix for
# Evolvable Safety Benchmarking: A Multi-Agent Pipeline for LVLMs

**LLM Clarification**  We note that large language models were used to aid in polishing the writing of this paper, but they were not involved in the research design, experimental process, or analysis.

**Ethics Statement**  This work on safety benchmarking for large language models (LLMs) is conducted in alignment with the ICLR Code of Ethics. Our research involves the systematic evaluation of AI models for potential harmful outputs; therefore, we have implemented strict ethical safeguards. All prompts used to elicit harmful behaviors are carefully curated from established safety benchmarks and are designed for measurement, not for generating or disseminating harmful content. The evaluated models are publicly available or used under their respective research licenses. We do not involve human subjects directly. However, we acknowledge the broader ethical implications: the benchmarks and insights generated could potentially be misused to circumvent safety mechanisms. To mitigate this, we focus on reporting aggregated metrics and trends rather than providing specific, exploitable adversarial prompts. Our goal is to contribute to the development of safer AI systems by providing rigorous, transparent, and responsible evaluation methodologies. We have no conflicts of interest to declare.

**Reproducibility Statement**  To ensure the reproducibility of our safety benchmarking study, we provide comprehensive details and resources. Code and data will be released after the paper is accepted. For the proprietary models we evaluate (e.g., GPT-4, Claude-3), we specify the exact API versions and the exhaustive set of inference parameters (e.g., temperature, max tokens). For open-source models, we provide the specific model repository URLs and hashes. The results for all experiments are presented with the mean and standard deviation across multiple runs.

## A  Related Work

### A.1  LVLM Safety Evaluation

Numerous safety evaluation benchmarks for large vision-language models (LVLMs) have been proposed. We categorize them into four types and describe them respectively.

**General Safety Benchmarks**  These benchmarks provide broad evaluations covering multiple safety aspects such as general safety, out-of-distribution generalization, and overall trustworthiness. MM-SafetyBench (Liu et al., 2024b) is an early benchmark containing 5,040 text-image pairs across 13 scenarios. MMDT (Xu et al., 2025) builds a unified platform for comprehensive safety evaluation, covering safety, hallucination, fairness, privacy, adversarial robustness, and OOD generalization. Similarly, MultiTrust (Zhang et al., 2024c), USB (Zheng et al., 2025), and MLLMGuard (Gu et al., 2024) are unified comprehensive benchmarks that cover sufficient aspects: truthfulness, safety, robustness, fairness, privacy, etc.. Unicorn (Tu et al., 2023)is a comprehensive benchmark evaluating Out-of-Distribution (OOD) generalization and adversarial robustness. AVIBench (Zhang et al., 2024a) focuses on evaluating the robustness against adversarial visual instructions.

**Jailbreak Benchmarks**  These datasets are specifically designed to assess the vulnerability of MLLMs to jailbreak attacks. JailBreakV-28K (Luo et al., 2024) is a widely used large-scale benchmark with 28,000 test cases, demonstrating high success rates for transferred LLM attacks. MMJ-Bench (Weng et al., 2024) is a unified pipeline for systematically evaluating jailbreak attacks and

defense techniques for MLLMs. Empirical studies on GPT-4o (Ying et al., 2024b) and broader landscape analyses (Wang et al., 2024a) also contribute to understanding jailbreaking vulnerabilities.

**Domain-Specific Benchmarks**  These benchmarks evaluate safety risks in specific contexts or using real-world data formats like memes and AI-generated content. MemeSafetyBench (Lee et al., 2025) contains 50,430 instances using real meme images to evaluate VLM safety. GOAT-Bench (Lin et al., 2024) comprises over 6K memes covering fine-grained themes like implicit hate speech and cyberbullying. For generated images, ExtremeAIGC (Chandna et al., 2025) benchmarks LMM vulnerability to AI-generated extremist content. UnsafeBench (Qu et al., 2024) evaluates image safety classifiers on real-world and AI-generated images, identifying performance degradation on AI-generated content. Other fields also introduce their benchmarks. SHIELD (Shi et al., 2024) evaluates MLLM capabilities in face spoofing and forgery detection across RGB, infrared, and depth modalities. Argus Inspection (Yao et al., 2025) evaluates visual fine-grained perception and causal reasoning capabilities. VLSBench (Hu et al., 2024) resolves visual leakage issues in multimodal safety evaluation. MOSSBench (Li et al., 2024c) evaluates oversensitivity in MLLMs, measuring refusal rates for harmless queries.

## A.2 AUTOMATED BENCHMARK (DATASET) CREATION

Large Language Models (LLMs) or LVLMs demonstrate remarkable capabilities in data generation, enabling their use for creating or updating evaluation datasets, thereby replacing labor-intensive manual data curation processes (Liu et al., 2024a). We categorize existing methods for generating evaluation benchmarks into three types:

**1) Dynamic Evaluation:** Methods like DyVal (Zhu et al., 2024a), DyVal 2 (Zhu et al., 2024b), DME (Yang et al., 2024), DARG (Zhang et al., 2024d), and LatestEval (Li et al., 2024d) propose to evaluate LLMs via modifying existing benchmarks. These methods aim to enhance dataset difficulty through dynamic updates and mitigate data contamination issues. For instance, Wang *et al.*propose a self-evolving method for optimizing test prompts (Wang et al., 2024b). SDEval proposes a dynamic safety evaluation framework for LVLMs, which can also be used for model performance evaluation (Wang et al., 2025).

**2) Benchmark Extension:** These methods are used to update or extend existing benchmarks. For instance, AutoBencher generates question-answer pairs by retrieving topic information from a database, optimizing existing benchmarks to improve their diversity and quality (Li et al., 2024b); EvoEval explores extending existing code benchmarks to different code domains through LLM-based augmentation methods and human verification (Xia et al., 2024). Furthermore, GSM-Symbolic (Mirzadeh et al., 2024) and GSM-Infinite (Zhou et al.) are extensions based on existing datasets like GSM-8K (Cobbe et al., 2021).

**3) Benchmark Construction:** AutoBench utilizes LLMs to annotate image-based question-answer pairs for evaluation (Qiu et al., 2024); TaskMeAnything generates input-output pairs based on question-answer templates for building customized multimodal evaluation data (Zhang et al., 2024b); BenchAgents combines agents with human collaboration for dataset construction (Butt et al., 2024); DataGen proposes a unified framework for dataset construction and data augmentation (Huang et al., 2024). AutoBench-V is the first agent-based vision-language automated evaluation framework (Bao et al., 2024). Additionally, LLM-as-an-Examiner (Bai et al., 2023), LLM-as-an-Interviewer (Kim et al., 2024), BenchBuilder (Li et al., 2024a), StructEval (Cao et al., 2024), TreeEval (Li et al., 2025), and DeepEval (Li et al., 2023) have also constructed various automated evaluation frameworks.

These methods advance automated evaluation technology to varying degrees. However, none of them establish an end-to-end LVLM safety benchmarking system. In contrast, we model and modularize the entire dataset construction pipeline, and are currently the only multi-agent system specifically designed for LVLM safety benchmarking.

## B  DATA COLLECTION

To construct a raw data pool for VLSafetyBencher, we aggregate data from four sources to guarantee comprehensiveness and diversity: existing safety datasets, general image datasets, synthetic images,

Table 1: The two-layer safety taxonomy and sample distribution. The total number of samples in the generated benchmark is 4,000.

| Category | Samples | Ratio(%) |
|---|---|---|
| **Privacy** | **426** | **10.65** |
| • Privacy Invasion | 206 | 5.15 |
| • Identity Misuse | 220 | 5.5 |
| **Bias** | **419** | **10.475** |
| • Discrimination | 179 | 4.475 |
| • Insulting & Harassing | 240 | 6.00 |
| **Toxicity** | **933** | **23.325** |
| • Deception | 113 | 2.825 |
| • Dark Themes | 120 | 3.00 |
| • Horror and Gore | 80 | 2.00 |
| • Pornograph | 121 | 3.025 |
| • Sexual Crimes | 65 | 1.625 |
| • Animal Abuse | 124 | 3.10 |
| • Dangerous Behavior | 310 | 7.75 |
| **Legality** | **584** | **14.60** |
| • Financial Fraud | 203 | 5.075 |
| • Terrorism | 181 | 4.525 |
| • Cybersecurity | 200 | 5.00 |
| **Misinformation** | **491** | **12.275** |
| • False Information | 207 | 5.175 |
| • Sensitive Information | 127 | 3.175 |
| • Public Misinformation | 157 | 3.925 |
| **Health Risk** | **1147** | **28.675** |
| • Health Hazards | 235 | 5.875 |
| • Environmental Harm | 139 | 3.475 |
| • Physical Harm | 382 | 9.55 |
| • Psychological Harm | 391 | 9.775 |
| **Total** | **4000** | **100** |

and social media data. We follow (Zhang et al., 2025) and leverage CLIP (Radford et al., 2021) to conduct coarse filtering, ensuring that all selected images contain potentially harmful information. After filtration, the data pool comprises around 300K images, with 134K derived from existing datasets, 20K from general images, 40K generated via diffusion models, and 106K scraped from social media. Below, we detail the information of each source.

- **Existing Safety Datasets**: We select 134K image-question pairs from open-source LVLM safety benchmarks. The involved datasets include SafeBench (Ying et al., 2024a), MLLM-Guard (Gu et al., 2024), BeaverTails-V (Ji et al., 2025), and SPA-VL (Zhang et al., 2025).

- **General Image Datasets**: We curate 20K risk images from LAION-5B (Schuhmann et al., 2022). We follow (Zhang et al., 2025) to utilize CLIP to identify harmful images. The LAION Safety Toolkit (Laion-AI, 2021) is also adopted to identify porn images.

- **Synthetic Images**: We also collect 40K synthesized images. Generative models such as Stable Diffusion 3 (Esser et al., 2024), FLUX.1 dev (Labs, 2024), and MidJourney v6 (Midjourney, 2023) are included. The generation process is guided by 20K harmful prompts generated by uncensored models. Diverse safety categories are covered.

- **Social Media Data**: We scrape 106K harmful images from the internet, including social media (Xiaohongshu[1] and X[2]) and news platform (Toutiao[3]). To ensure ethical compliance, we filter out images containing personally identifiable information (PII). In addition, we perform watermark removal on select images.

---

[1]https://www.xiaohongshu.com

[2]https://x.com

[3]https://www.toutiao.com

Table 2: Safety evaluation of 35 mainstream LVLMs. In addition to ASR, we also present safety rate (SR), where SR=1-ASR. **Blod** indicates the best and underline indicates the second.

| Model | Privacy | Bias | Toxic | Legal | Misinfo | Health | ASR | SR |
|---|---|---|---|---|---|---|---|---|
| Ovis2-8B | 51.59 | 34.04 | 43.87 | 51.70 | 42.35 | 46.16 | 44.85 | 55.15 |
| Ovis2-16B | 45.86 | 39.72 | 43.68 | 52.13 | 31.12 | 42.37 | 42.73 | 57.27 |
| Ovis2-34B | 45.86 | 39.01 | 42.34 | 45.11 | 31.63 | 37.50 | 40.40 | 59.60 |
| SAIL-VL-8B | 57.32 | 42.55 | 51.70 | 61.49 | 43.88 | 52.49 | 51.70 | 48.30 |
| MiniCPM-V2.6 | 51.59 | 43.97 | 46.16 | 57.66 | 41.84 | 50.25 | 48.23 | 51.77 |
| QVQ-72B-Preview | 38.56 | 32.62 | 33.61 | 38.65 | 40.00 | 38.24 | 36.35 | 63.65 |
| Qwen2.5-VL-32B-Instruct | 45.39 | 17.73 | 28.36 | 45.93 | 15.90 | 36.44 | 31.44 | 68.56 |
| Qwen2.5-VL-72B-Instruct | 30.77 | 6.12 | 26.41 | 29.08 | 16.22 | 20.19 | 22.42 | 77.58 |
| Mistral-Small-3.1-24B-Instruct | 31.21 | 10.64 | 22.77 | 31.91 | 15.82 | 19.69 | 21.92 | 78.08 |
| Pixtral-12B-2409 | 57.96 | 39.01 | 52.77 | 58.72 | 44.39 | 46.81 | 50.22 | 49.78 |
| InstructBLIP-Vicuna-7B | 77.71 | 68.79 | 70.01 | 78.94 | 68.88 | 67.73 | 72.19 | 27.81 |
| InstructBLIP-Vicuna-13B | 77.78 | 71.43 | 65.48 | 78.46 | 70.27 | 68.35 | 70.12 | 29.88 |
| Phi-3.5-Vision-Instruct | 9.55 | 42.55 | 14.41 | 7.23 | 41.33 | 13.96 | 19.09 | 80.91 |
| Kimi-VL-A3B-Instruct | 57.32 | 48.23 | 46.44 | 52.13 | 42.35 | 52.78 | 49.43 | 50.57 |
| InternVL3-1B-Instruct | 50.00 | 50.00 | 59.78 | 67.27 | 33.33 | 63.25 | 56.27 | 43.73 |
| InternVL3-9B-Instruct | 45.22 | 46.10 | 45.17 | 53.19 | 36.22 | 51.47 | 46.58 | 53.42 |
| InternVL3-38B-Instruct | 37.58 | 36.17 | 35.55 | 43.62 | 31.63 | 37.94 | 36.84 | 63.16 |
| InternVL3-78B-Instruct | 33.12 | 30.50 | 29.45 | 34.68 | 24.49 | 32.81 | 30.75 | 69.25 |
| Gemma-3-4B-It | 35.67 | 14.89 | 20.52 | 34.26 | 17.86 | 17.53 | 22.14 | 77.86 |
| Gemma-3-27B-It | 36.31 | 9.93 | 16.07 | 29.15 | 16.33 | 17.58 | 19.45 | 80.55 |
| DeepSeek-VL2-Tiny | 62.42 | 73.05 | 60.78 | 60.85 | 67.35 | 60.89 | 63.09 | 36.91 |
| DeepSeek-VL2 | 48.41 | 79.43 | 43.72 | 46.38 | 69.90 | 52.70 | 53.41 | 46.59 |
| DeepSeek-VL2-Small | 52.23 | 81.56 | 55.00 | 51.28 | 76.02 | 56.48 | 59.89 | 40.11 |
| GLM-4.1V-9B-Thinking | 69.43 | 45.39 | 54.10 | 68.30 | 46.94 | 62.59 | 57.50 | 42.50 |
| LLaVA-V1.6-Vicuna-13B | 59.87 | 30.50 | 42.47 | 56.60 | 33.16 | 45.00 | 44.17 | 55.83 |
| LLaVA-V1.6-Mistral-7B | 64.97 | 26.95 | 53.72 | 65.32 | 33.67 | 55.63 | 51.48 | 48.52 |
| LLaVA-Next-110B | 59.24 | 52.48 | 52.05 | 58.94 | 47.45 | 48.44 | 52.35 | 47.65 |
| LLaMA-3.2-11B-Vision-Instruct | 36.31 | 34.75 | 31.87 | 31.70 | 33.67 | 25.05 | 31.35 | 68.65 |
| Grok-4 | 50.94 | 35.48 | 29.80 | 30.53 | 23.75 | 38.34 | 34.09 | 65.91 |
| GPT-4o | 7.33 | 12.50 | 5.69 | 2.95 | 6.91 | 4.62 | 6.22 | 93.78 |
| GPT-4.1 | 8.05 | 12.50 | 7.30 | 4.03 | 6.84 | 5.44 | 7.13 | 92.87 |
| Qwen-VL-Max | 28.48 | 17.39 | 19.58 | 26.27 | 11.23 | 22.22 | 20.73 | 79.27 |
| Gemini-2.5-Pro | 37.50 | 19.82 | 22.41 | 20.79 | 18.47 | 14.43 | 21.41 | 78.59 |
| **Claude-Sonnet-4** | **1.32** | **3.55** | **1.88** | **1.55** | **4.15** | **1.89** | **2.22** | **97.78** |

## C    SAFETY TAXONOMY

We integrate the taxonomies of (Gu et al., 2024) and (Ji et al., 2025) to define a two-layer tree comprising 6 categories and 20 subcategories. The 6 categories are: *Privacy*, *Bias*, *Toxicity*, *Legality*, *Misinformation*, and *Health Risk*. The subcategories and the number of samples of each (sub)category is presented in Table 1.

## D    LVLM SAFETY EVALUATION

Using the benchmark generated by VLSafetyBencher, we evaluate 35 mainstream LVLMs in Table 2. We can observe significant safety disparity, where the safety rate ranges from 27.81% (InstructBLIP-Vicuna-7B) to 97.78% (Claude-Sonnet-4 ). It is observed that safety shows a general upward trend with increasing model scale within the same series, which is corroborated by the Qwen2.5-VL and InternVL3 model families.