# OpenReview forum: "Evolvable Safety Benchmarking: Multi-agent Pipeline for LVLMs"
_ICLR.cc/2026/Conference — ICLR 2026 Conference Withdrawn Submission_

### Official Review · Reviewer_pLqf · 2025-10-16

**Soundness:** 2
**Presentation:** 3
**Contribution:** 3
**Rating:** 4
**Confidence:** 4

**Summary:**

The authors present VLSafetyBencher, an automated and evolvable safety benchmarking framework for Large Vision-Language Models. The system addresses limitations of existing benchmarks such as static datasets and manual annotation by employing a multi-agent pipeline that automatically generates, optimizes, and updates multimodal safety data. The framework includes four agents: a Preprocessing Agent that cleans and categorizes data into six risk types; a Cross-Modal Agent that creates image–question pairs following principles of modality dependency, complementarity, and conflict; an Augmentation Agent that diversifies and strengthens samples via paraphrasing and jailbreak-like transformations; and a Post-Sampling Agent that selects optimal samples by maximizing separability, harmfulness, and diversity. Benchmark quality is assessed through four metrics. Experiments show that VLSafetyBencher outperforms prior benchmarks like SafeBench and AutoBencher, achieving higher separability and diversity, while remaining fully automated and inexpensive. Evaluating 35 LVLMs reveals significant safety variation closed-source models like GPT-4o and Claude-Sonnet-4 achieve over 90% safety rates, while open-source models lag behind.

**Strengths:**

1. The authors introduce a novel and fully automated framework for LVLM safety benchmarking, which significantly reduces human effort and enables continuous, scalable evaluation as models evolve. This automation represents a clear advancement over existing static, manually curated benchmarks.

2. The system is highly efficient and cost-effective, completing the entire benchmark generation process within a week at an extremely low cost (around $1.34) using only standard API calls. This makes it practical for iterative or large-scale deployment.

3. The framework’s design for evolvability is conceptually impactful. By allowing dynamic regeneration and replacement of benchmark samples, it enables long-term adaptability to new threat patterns, model architectures, and emerging safety concerns.

**Weaknesses:**

1. The framework poses a clear misuse risk since it can be repurposed to automatically generate large-scale harmful or adversarial content. While the authors emphasize safety benchmarking, they do not discuss any ethical safeguards, access control, or mitigation strategies to prevent malicious exploitation. This omission raises serious concerns about dual-use potential.

2. The system heavily relies on LLM-based judgments for labeling and filtering, which may inherit or amplify the biases, inconsistencies, and hallucinations of the underlying models. Without human validation or bias correction, the resulting benchmarks risk reflecting model-specific moral or cultural biases rather than objective safety standards.

3. The proposed evaluation metrics (MAD, MEAN, GAP, DIV) are intuitive but heuristic, lacking strong theoretical justification or empirical validation. The weighting parameters (α, β, γ) are manually tuned, and there is no adaptive or data-driven mechanism to optimize them.

4. The reliance on closed-source APIs such as GPT-4o and Claude limits reproducibility and transparency, as other researchers cannot replicate results under identical conditions.

5. The authors present a fully automated pipeline but omit any human evaluation or expert verification, which undermines the reliability of the generated benchmark and its alignment with human safety norms.

**Questions:**

1. The framework uses LLMs to assess harmfulness and safety. How do the authors ensure that the model used for labeling does not bias the benchmark toward its own alignment characteristics? Have they tested whether using a different evaluator model changes the benchmark ranking?

2. The Cross-Modal Agent is said to construct samples under three strategies modality dependency, complementarity, and conflict. Can the authors provide quantitative statistics on how many samples fall under each type, and whether any one type dominates the final benchmark composition?

3. The benchmark quality metrics (MAD, MEAN, GAP, DIV) are primarily statistical. Was there any human evaluation or expert audit conducted to verify that the benchmark samples actually reflect real-world safety risks or harmful content?

**Details Of Ethics Concerns:**

The proposed framework can automatically generate, manipulate, and optimize harmful or adversarial multimodal content as part of its benchmarking process. While the authors intend this for safety evaluation, the same methodology could be repurposed to mass-produce unsafe, toxic, or misleading data, effectively serving as a harmful content generator. Moreover, the paper does not discuss access control, misuse prevention, or responsible release strategies, leaving open the risk of unethical or malicious application. In short, the framework provides valuable safety insights but also introduces dual-use risks that warrant careful ethical consideration and oversight.

---

### Official Review · Reviewer_XLQD · 2025-10-18

**Soundness:** 3
**Presentation:** 3
**Contribution:** 3
**Rating:** 6
**Confidence:** 4

**Summary:**

This paper utilizes multi-agent pipeline to synthesize harmful image-question pair and introduces the optimization-based sampling method to select high-value test data. Various experiments confirm the effectiveness of proposed method.

**Strengths:**

The framework is clear and easy to follow specific agent operation.

**Weaknesses:**

This paper has several weaknesses.

First, as stated in the Framework, the Cross-Modal Agent groups samples into Modality Dependency, Complementarity, and Conflict. It is unclear whether this classification is 100% accurate. If some cases are misclassified, what consequences would these misclassified samples bring? If the classification is perfect, does that imply a super-powerful agent?

Second, the Augmentation Agent includes various text and image augmentations. The authors should release which augmentation is most effective and provide rationale analysis. This would strengthen the paper’s solidity.

Third, for the POST-SAMPLING AGENT, the authors use three metrics: Separability, Harmfulness, and Diversity. However, the authors should rethink why these three metrics are chosen over others. Additionally, regarding the optimization weights for these three modules, Figure 3 does not clearly report the optimal weight combination. While the authors provide some discussion in Line 460, it is insufficient.

In summary, I acknowledge that this paper is interesting and the authors have conducted various evaluation experiments. However, the rationale for the method and the discussion of ablation studies remain important.

**Questions:**

Refer to weakness.

---

### Official Review · Reviewer_aTJA · 2025-10-25

**Soundness:** 2
**Presentation:** 3
**Contribution:** 2
**Rating:** 4
**Confidence:** 5

**Summary:**

This paper proposes VLSafetyBencher, a fully automated multi-agent system for LVLM safety benchmarking, addressing high costs, staticity, and weak discriminative power of existing benchmarks. It uses four collaborative agents (Preprocessing, Cross-Modal, Augmentation, Post-Sampling) and an optimization-based sampling algorithm to select high-quality data. Experiments show it constructs benchmarks efficiently (1 week, $1.34 cost), outperforms baselines in metrics like GAP (≈70%), and effectively evaluates 35 LVLMs, revealing large safety disparities. Code and data will be released post-acceptance.

**Strengths:**

First, it automates LVLM safety benchmarking via 4 collaborative agents, cutting construction time to ~1 week and cost to $1.34, far less than manual methods.

Second, its optimization-based sampling boosts discriminative power, showing a 70% safety rate gap between top and bottom LVLMs, outperforming baselines.

Third, it enables dynamic benchmark updates, enhancing existing ones (e.g., SafeBench) by replacing low-efficiency samples to improve metrics.

**Weaknesses:**

First, The paper has relatively low innovation, as its multi-agent framework and sampling methods build on existing automated benchmarking ideas.

Second, It fails to detail the agent system construction, with no supplementary info even in the appendix.

**Questions:**

see Weaknesses

---

### Official Review · Reviewer_nzLn · 2025-11-01

**Soundness:** 3
**Presentation:** 3
**Contribution:** 3
**Rating:** 4
**Confidence:** 3

**Summary:**

This paper presents VLSafetyBencher, a multi-agent system designed to automate the construction and updating of safety benchmarks for large vision-language models (LVLMs). The framework integrates four agents—Preprocessing, Cross-Modal, Augmentation, and Post-Sampling—to perform data cleaning, harmful multimodal pair generation, augmentation, and optimization-based sampling. The authors formalize benchmark desiderata (separability, harmfulness, diversity) and propose an iterative selection algorithm to optimize these criteria. Experiments show that the generated benchmarks outperform prior works such as SafeBench, MLLMGuard, and AutoBencher, achieving greater discriminative power and construction efficiency.

**Strengths:**

- Trendy and practically meaningful topic on LVLM safety evaluation
- Clear presentation and strong experimental validation
- Automated pipeline reduces human cost and enables benchmark updating
- Comprehensive ablation analysis and comparison with prior benchmarks

**Weaknesses:**

- Limited technical novelty
- Dataset construction relies on existing collected data

**Questions:**

The paper proposes a well-structured and practically useful system for automating LVLM safety benchmark construction. The design of multiple agents and the optimization-based selection pipeline are clearly presented and empirically validated.

However, two major concerns limit the impact of this work at a top-tier venue:

First, compared with prior agent-based benchmark construction frameworks such as AutoBencher (Li et al., 2024), the claimed innovation of this paper is not sufficiently substantial. AutoBencher already formulates benchmark generation as an automated agent pipeline, the proposed VLSafetyBencher essentially inherits the same high-level agent-based paradigm, with the main modification being the shift from textual to multimodal (LVLM) data and the transition from dataset-level optimization to sample-level. However, this change mainly reflects a granularity adjustment, rather than a conceptual or algorithmic breakthrough.

Second, the dataset construction of VLSafetyBencher heavily depends on existing collected or filtered data sources rather than generating novel, risk-evolving, or task-driven safety samples.

---

### Note · Authors · 2025-11-17

I have read and agree with the venue's withdrawal policy on behalf of myself and my co-authors.